# Variability and combination as ensemble of mineral dust forecast during the 2021 CADDIWA experiment, using the WRF 3.7.1 and CHIMERE v2020r3 models.

Laurent MENUT[1]

[1]Laboratoire de Météorologie Dynamique (LMD), Ecole Polytechnique, IPSL Research University, Ecole Normale Supérieure, Université Paris-Saclay, Sorbonne Universités, UPMC Université Paris 06, CNRS, Route de Saclay, 91128 Palaiseau, France

**Correspondence:** Laurent Menut, laurent.menut@lmd.ipsl.fr

**Abstract.** As an operational support to define the CADDIWA field campaign which took place in the Cape Verde area, the coupled regional model WRF-CHIMERE is deployed in forecast mode during the summer 2021. The simulation domain covers the West Africa and the East Atlantic and allows the modeling of dust emissions and their transport to the Atlantic. On this route, we find Cape Verde which was used as a base for measurements during the CADDIWA campaign. Meteorological variables and mineral dust concentrations are forecasted on a horizontal grid with a 30 km resolution and from the surface to 200 hPa. For a given day D, simulations are initialized from D-1 analyses and run four days until D+4, yielding up to six available simulations on a given day. For each day, we thus have six different calculations, with expectantly a better precision the closer we get to the analysis (lead D-1). In this study, a quantification of the forecast variability of wind, temperature, precipitations and mineral dust concentrations according to the modelled lead is presented. It is shown that the forecast quality does not decrease with time and the high variability observed on some days for some variables (wind, temperature) does not explain the behavior of other dependent and downwind variables (mineral dust concentrations). A new method is also tested to create an ensemble without perturbing input data, but considering six forecast leads available for each date as members of an ensemble forecast. It has been shown that this new forecast based on this ensemble, is able to give better results for two AErosol RObotic NETwork (AERONET) stations on the four available for Aerosol Optical Depth observations. This could open the door to further testing with more complex operational systems.

## 1 Introduction

Over the western Africa, and during the boreal summer, mesoscale convective systems are moving from East to West and interact with the African easterly jet (AEJ), African easterly waves (AEWs) and mineral dust plumes, (Knippertz and Todd, 2010; Marsham et al., 2011; Cuesta et al., 2020). Leaving the African continent to arrive above the Atlantic Ocean, they can generate tropical storms. The magnitude of interactions between these systems and the mineral dust concentrations via the direct and indirect effects of aerosols on meteorology remain unclear, (Lavaysse et al., 2011; Price et al., 2018; Martinez and Chaboureau, 2018). This motivated the deployment of the Clouds-Atmospheric Dynamics-Dust Interactions in West Africa

(CADDIWA) field campaign, (Flamant et al., 2022). The measurements include long-term surface stations and dedicated airborne measurements. Aircraft were located at Sal (Cape Verde), under the wind flow coming from western Africa. In addition to the local study of storms generation, these aircraft measurements were also designed to help with the validation of space-borne wind and aerosol products as those of Aeolus, EarthCare and IASI satellites missions, (Clerbaux et al., 2009; Illingworth et al., 2015; Martin et al., 2021). In addition to these measurements, numerical modelling is performed with the coupled regional model WRF-CHIMERE. Simulations are initialized at D-1, in order to provide forecasts from D+0 to D+4.

Before studying the interaction between aerosols, clouds and radiation using a numerical tool, it is important to assess its accuracy. Forecast is a useful tool for this kind of evaluation. Simulating the same day several times with a different meteorology is a way to quantify the model variability. It is close to an ensemble simulation, even if the number of members is lower, (Atger, 1999; Toth et al., 2001; Richardson, 2001). By comparing the six days of forecast simulations, ranging from D-1 to D+4, but for a given date enables to quantify the variability of the model.

Another aspect will be analyzed in this study: as several forecast leads correspond to the simulation of the same period but with different initial conditions for the meteorology, we can imagine that the leads are equivalent to ensemble modeling members. Ensemble modelling is widely used in forecast of meteorology or air quality, (Delle Monache et al., 2006), (Vautard, 2006), (Benedetti et al., 2018). But in general, the ensemble is built using the same model with perturbations. Some other techniques exist such as the "poor man's ensemble" and are widely used in meteorology, (Ebert, 2001), (Buizza et al., 2003), (Bowler et al., 2008). To our knowledge, these approaches are not used for Chemistry-Transport Modelling (CTM). They consist in using different models but making the forecast of the same period. Also in meteorology, they can be used in operational centers to update the covariance matrixes used for the data assimilation.

In this study, we aim to answer the following question: *Is it possible to use several forecast leads as if it was an ensemble and improve the quality of the forecast?* If the result is positive, it means it is possible to run less ensemble simulations each day, hence a faster forecast, while still improving the quality of the forecast. And for institutes which do not have sufficient computing resources to perform classical ensemble simulations, it still allows them to have a probabilistic approach on their forecast based on a single model.

The main goal of this study is thus first to quantify the variability on temperature, wind, precipitation rates, Aerosol Optical Depth (AOD) and surface concentration of mineral dust as a function of the forecast lead time. Second to try to establish some correlations between possible differences in forecast results. With this quantification, we can assess the robustness of the forecast and the degree of confidence available that experimenters may have during field airborne campaign such as CADDIWA. Section 2 presents the modelling system and the studied period. Section 3 presents the results of the comparison between the forecast with different lead times. Section 4 presents a tentative approach of mixing several leads for the same day in order to mimic an ensemble forecast. Results are compared to AERONET Aerosol Optical Depth measurements. Section 5 presents conclusions of the study.

## 2 The modeling system

### 2.1 The modelling tools

In this study, we use the WRF-CHIMERE model built with WRF 3.7.1, (Powers et al., 2017), and CHIMERE 2020r3, (Menut et al., 2021). These two models are coupled using the OASIS3-MCT external coupler, (Craig et al., 2017). The WRF model is forced with the global scale forecast fields from NCEP Global Forecast System (GFS), (Halperin et al., 2020). For this experiment, a specific configuration was designed in order to have a lower numerical cost. Indeed, the goal was to launch the simulation for six days, from (D-1), the day before the current date to (D+4) four days in advance. This long forecast was designed to allow to the aircraft scientists to have enough time to decide what flight plan to use, depending on the meteorological situation to come. Daily simulations were launched at midnight to benefit from the latest forecast meteorological field and needed to be available to scientists by 8 am local time in Cape Verde (0500 UTC), including all post-processed figures. These constraints of real-time forecast led to have a light version of the model where only mineral dust are modelled. The model is also used in offline mode, meaning that there is no feedbacks of aerosols on meteorology, in order to ensure the stability of the calculation. Mineral dust are modelled with ten bins from 0.01 to 40 $\mu$m. The dust emissions scheme used is the one of Alfaro and Gomes (2001), modified by Menut et al. (2005). Note that the CHIMERE model is also used daily in forecast mode for air quality with all available chemical processes, being operated by operational centers such as Prevair or Copernicus, (Rouïl et al., 2009; Marécal et al., 2015).

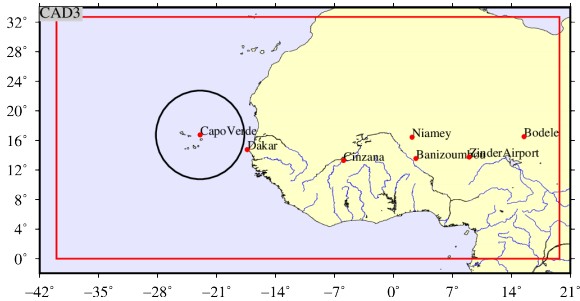

**Figure 1.** *Model domain in red, with the main studied locations: from East to West: Bodélé, Zinder, Banizoumbou, Niamey, Cinzana, Dakar, Cape Verde. The black circle is the possible range of aircraft measurements during the campaign.*

The model domain, Figure 1, is defined with the same horizontal grid for WRF and CHIMERE and covers a part of the Atlantic Ocean and West Africa, from -40$^o$E to +20$^o$E in longitude and 0$^o$ to 33$^o$N in latitude. It is constituted of 200 $\times$ 110 cells with a constant resolution of 30 km. The WRF model has 32 vertical levels from the surface to 50 hPa. CHIMERE has less vertical levels with 15 layers from the surface to 300 hPa. This domain was designed to be able to model at the same time: (i) mineral dust emissions in Africa, from Dakar to Bodélé, (ii) model transport from Africa to the Atlantic Ocean, (iii) have the measurement site of Cape Verde not too close to the domain boundaries. In Figure 1, the locations of Cape Verde, Dakar, Bodélé, Zinder, Niamey, Banizoumbou, Cinzana are reported. Model results were daily extracted at these locations for the

CADDIWA scientists based in Cape Verde with the aircrafts. The circle indicates the possible range of aircraft measurements during the campaign around the Island of Sal.

## 2.2 The observations

The goal is not to perform a comparison of model with observations. It will be done only at the end of the study with a comparison of measured versus modeled Aerosol Optical Depth (AOD) and for the 2m temperature and 10m wind speed for some locations.

For the AOD, The AErosol RObotic NETwork global remote sensing network (AERONET, https://aeronet.gsfc.nasa.gov/)
level 1.5 measurements are used, (Holben et al., 2001). The AOD at a wavelength of $\lambda$=675 nm are daily averaged and compared to daily averaged modelled values. For the meteorological variables, the measurements provided by the Weather Information website of the University of Wyoming (UWYO) are used (http://www.weather.uwyo.edu/). Data are provided for 2m temperature and 10m wind speed. It is noticeable that the data are delivered as integer values, restraining the accuracy of the comparison to the model results.

| Station | $\lambda$ | $\phi$ | AERONET | UWYO |
|---------|-----------|--------|---------|------|
| Name | (°E) | (°N) | | |
| Bodélé (Tchad) | 15.5 | 16.5 | x | |
| Zinder (Niger) | 8.98 | 13.75 | x | x |
| Banizoumbou (Niger) | 2.66 | 13.54 | x | |
| Niamey (Niger) | 2.2 | 16.43 | x | x |
| Cinzana (Mali) | -5.93 | 13.28 | x | |
| Dakar (Senegal) | -17.36 | 14.75 | x | x |
| CapeVerde (Cabo Verde) | -22.95 | 16.75 | x | x |

**Table 1.** *List of the AERONET and meteorological UWYO sites used for the comparisons between measured and modelled AOD, 2m temperature and 10m wind speed. Informations are the longitude $\lambda$, latitude $\phi$ for each site.*

## 2.3 The modelled period and the Intensive Observation Periods

The field campaign was carried out from 8 to 21 September 2021, with airborne measurements around Sal Island in Cape Verde, (Flamant et al., 2022). In order to have a tested and robust forecast modelling system, the daily forecast started August 10, 2021 and ended November 1, 2021. Among all observations periods, two events were observed: the tropical perturbation called Pierre-Henri, passing south of Sal on September 11 and the period from 17 to 24 September with the passage over Sal
of the two Tropical Cyclones, called Peter and Rose. In this study, the results will be presented over two periods:

– section 3: only for the period September 1 to 30, 2021 for the part about the variability of the forecast during the CADDIWA field experiment.

– section 4: over the whole modelled period, for the period August 10 to November 1, 2021, for the part about the merging of the several forecast leads of modelled fields.

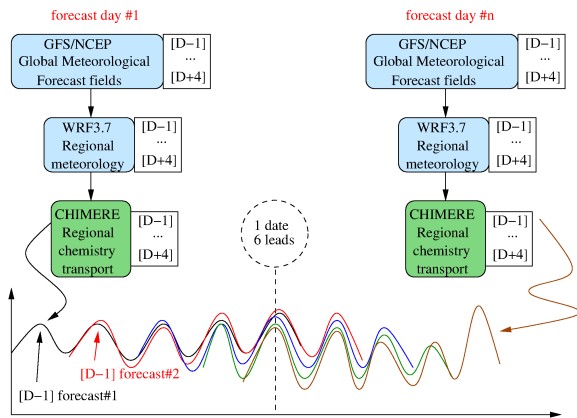

**Figure 2.** *Principle of the modelling system in forecast mode. Each day, the global meteorological fields are downloaded to force the regional WRF model. These regional fields are used to drive the CHIMERE chemistry-transport model, mainly for mineral dust emissions, transport and deposition. The procedure is repeated every day.*

For the results presentation, there are several possibilities. As presented in Figure 2, each day, the modelling system runs to simulate 6 days, from (D-1), the day before, to (D+4) four days in advance. With all these simulations, results may be discussed following two ways:

1. Comparison of all leads for one date: For example, for the September 11 at 16:00 UTC, we can display the result of the simulations performed:

   - September 12, forecast hour -8, (D-1)
   - September 11, forecast hour 16, (D+0)
   - September 10, forecast hour 16+24=40, (D+1)
   - September 9, forecast hour 40+24=64, (D+2)
   - September 8, forecast hour 64+24=88, (D+3)
   - September 7, forecast hour 88+24=112, (D+4)

   This comparison may be achieved with maps and vertical cross-sections.

2. Comparison between leads during the whole period: It is possible to build time-series using the (D-1) for all days, the (D+0) for all days until (D+4) for all days. In this case, we can calculate statistical scores between the time-series as if they were different model realizations.

In the following sections and the Appendix, when analysis consists in maps or vertical cross-sections, we selected the 11 September 2021 to present the results, being the day when the "Pierre-Henri" tropical perturbation was diagnosed above Sal, in Cape Verde Island, (Flamant et al., 2022).

## 3   Variability of forecast leads during CADDIWA experiment

Results are presented as statistical scores (defined in Section 3.3). They are calculated for data over Bodélé and Cape Verde. The
main goal being to compare the simulation leads and evaluate the variability from one lead to the next, there is no measurements in the analysis but only model versus model. The initialization of the model being performed using analyzed meteorological fields, (Halperin et al., 2020), the simulation of (D-1) is considered as the reference.

### 3.1   Time-series of surface mineral dust concentrations

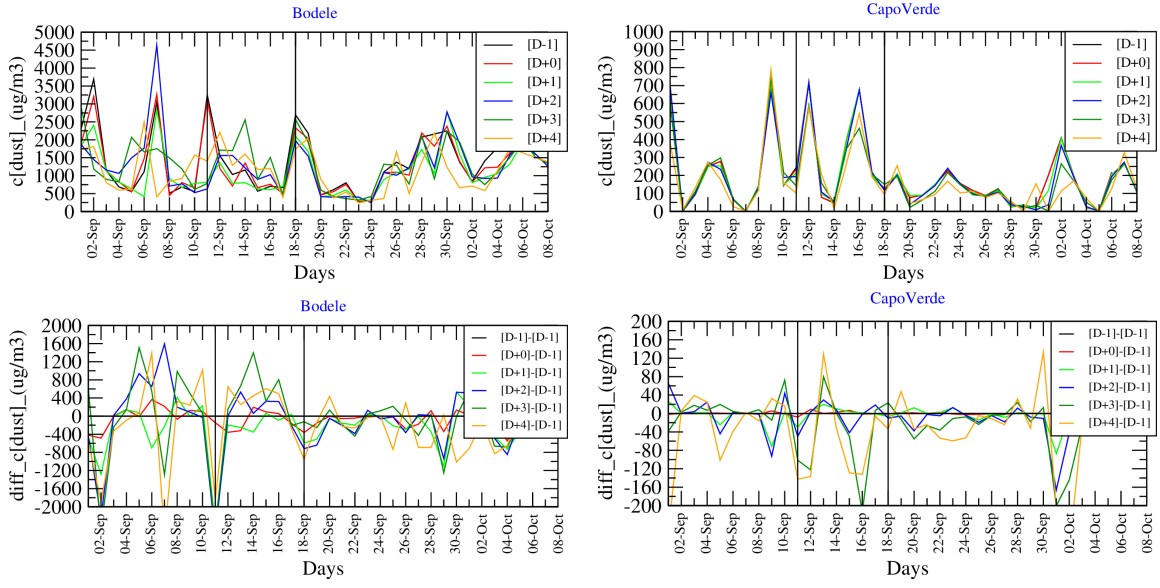

**Figure 3.** *Time-series of surface mineral dust concentrations ($\mu g.m^{-3}$) for each lead and of differences between leads.*

    Time-series are presented for two sites, Bodélé and Cape Verde. They are located on a Sahelian iso-latitude transect and
often used to quantify the amount of mineral dust emitted in the Saharan desert and after long-range transport of these dust, respectively, (Marticorena et al., 2010). Figure 3 presents time-series for the surface mineral dust concentrations ($\mu g.m^{-3}$). The variability of the forecast for these concentrations should be the result of a mix between the variabilities calculated with the 10m wind speed (an important parameter for the emissions) and the precipitation (a major sink). In Bodélé, the surface concentrations varies a lot between 0 and 5000 $\mu g.m^{-3}$. It seems huge but it is classical when being just over the main Saharan
source. The mass is composed of a large mass distribution and the major part of big particles are deposited before being

transported, close to the source. The variability from one lead to another is important and illustrates the impact of the wind speed variability. The most important differences are between -2000 and +2000 $\mu$g.m$^{-3}$ in Bodélé. A major forecast underestimation is calculated on the 11 September with -2000 $\mu$g.m$^{-3}$, meaning that an important peak of surface concentrations was modelled for (D-1) and (D+0) but not in advance. In Cape Verde, the surface concentrations are lower, logically after long-range transport

and because this site is not a mineral dust emissions hot spot. The concentrations remain high with peaks around 700 $\mu$g.m$^{-3}$. It is not the case of 11 September, but peaks are noted on 9, 12 and 16 September mainly. As well as for Bodélé, a large variability is calculated for 11 September, with forecast differences up to -200 $\mu$g.m$^{-3}$. As for Bodélé, the model underestimates the concentrations when the forecast is in advance. In addition to these results, time-series of 2m temperature, 10m wind speed and precipitation are presented in Appendix A1.

### 3.2 Maps of mineral dust concentrations and AOD

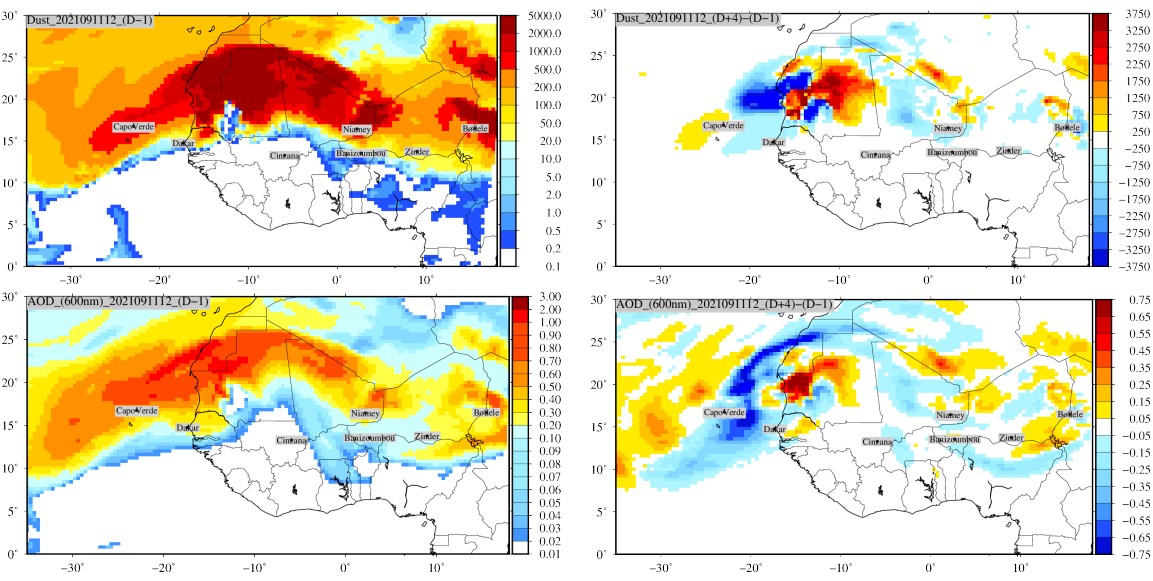

**Figure 4.** *Maps of surface mineral dust concentrations ($\mu$g.m$^{-3}$) and Aerosol Optical Depth for the 11 September at 12:00 UTC. Values are displayed (left) for the forecast lead (D-1) and (right) for the differences between the forecast leads (D-4)-(D-1).*

Maps are presented for surface mineral dust concentrations ($\mu$g.m$^{-3}$) and Aerosol Optical Depth for the 11 September at 12:00 UTC, in Figure 4. Note that complementary maps for wind speed and precipitation are presented in Appendix A2. The simulation shows a large mineral dust plume, flowing from Africa to northern Atlantic sea. The site of Cape Verde is under this plume and the trajectory over the ocean corresponds to the low wind speed values. The differences show the same kind

of dipole as diagnosed for the precipitation (Figure A4), showing that the shift between the forecast leads directly impacts the surface concentrations. With large positive values over land and negative over the sea, it is noticeable that the more recent forecast (D-1) diagnose a larger wind speed then a faster transport: the dust plume is more over land for (D+4), but is already

arrived over sea in (D-1). It means that over Cape Verde, the last forecast diagnosed finally more dust concentrations than the previous forecasts. The Aerosol Optical Depth represents well the behaviour of the mineral dust concentration, even if it diagnoses the radiative effect of all aerosols in the whole atmospheric column. The shape of the plume is slightly different and a larger spatial spread of the differences between the forecast leads is seen. The differences remain important in absolute values since they can reach $\pm 0.75$ when the maximum of AOD is 2. The variability in the forecast is important and show for this day that the forecast of (D+4) underestimated AOD over Cape Verde compared to (D-1). In addition to these horizontal maps, the same variables are analyzed in Appendix A3 as vertical cross-sections.

## 3.3 Statistical scores

Usually, the variables $O_t$ and $M_t$ stand for the observed and modeled values, respectively, at time $t$. In case of this study, as we want to quantify the variability of the forecast, the variable $O_t$ is the model realization at (D-1) and the variable $M_t$ is the model realization at leads (D+0) to (D+4). The mean value $\overline{X_N}$ is calculated as:

$$\overline{X_N} = \frac{1}{N} \sum_{t=1}^{N} X_t \tag{1}$$

with $N$ the total number of hours of the simulation. To quantify the temporal variability, the Pearson product moment correlation coefficient $R$ is calculated as:

$$R = \frac{\frac{1}{N} \sum_{t=1}^{N} (M_t - \overline{M_t}) \times (O_t - \overline{O_t})}{\sqrt{\frac{1}{N} \sum_{t=1}^{N} (M_t - \overline{M_t})^2 \times \frac{1}{N} \sum_{t=1}^{N} (O_t - \overline{O_t})^2}}, \tag{2}$$

The spatial correlation, noted R$_s$, uses the same formula type except it is calculated from the temporal mean averaged values of observations and model for each location where observations are available.

$$R_s = \frac{\sum_{i=1}^{I} (\overline{M_i} - \overline{\overline{M}})(\overline{O_i} - \overline{\overline{O}})}{\sqrt{\sum_{i=1}^{I} (\overline{M_i} - \overline{\overline{M}})^2 \sum_{i=1}^{I} (\overline{O_i} - \overline{\overline{O}})^2}} \tag{3}$$

where $I$ is the number of stations. The Root Mean Square Error (RMSE), is expressed as:

$$\text{RMSE} = \sqrt{\frac{1}{T} \sum_{t=1}^{T} (O_{t,i} - M_{t,i})^2} \tag{4}$$

To quantify the mean differences between the several leads, the bias is also quantified as:

$$\text{bias} = \frac{1}{N} \sum_{t=1}^{N} (M_t - O_t) \tag{5}$$

| | 10m wind speed (m.s$^{-1}$) | | | 2m temperature ($^o$C) | | |
|---|---|---|---|---|---|---|
| Lead | Mean | Bias | R | Mean | Bias | R |
| *Bodélé* | | | | | | |
| (D-1) | 3.89 | | | 32.2 | | |
| (D+0) | 3.83 | -0.060 | 0.23 | 32.1 | -0.082 | 0.90 |
| (D+1) | 3.72 | -0.161 | -0.17 | 32.2 | -0.013 | 0.80 |
| (D+2) | 3.78 | -0.110 | 0.06 | 32.2 | -0.023 | 0.79 |
| (D+3) | 3.83 | -0.056 | 0.05 | 32.0 | -0.145 | 0.75 |
| (D+4) | 3.69 | -0.199 | -0.04 | 31.9 | -0.305 | 0.79 |
| *Cape Verde* | | | | | | |
| (D-1) | 5.49 | | | 24.6 | | |
| (D+0) | 5.56 | 0.069 | 0.04 | 24.6 | 0.007 | 0.57 |
| (D+1) | 5.55 | 0.056 | -0.27 | 24.6 | 0.021 | 0.16 |
| (D+2) | 5.47 | -0.021 | -0.15 | 24.6 | 0.032 | 0.08 |
| (D+3) | 5.54 | 0.048 | 0.03 | 24.6 | 0.051 | 0.24 |
| (D+4) | 5.38 | -0.110 | -0.01 | 24.6 | 0.036 | 0.29 |

**Table 2.** *Statistical scores for the daily averaged 10-m wind-speed (m.s$^{-1}$) and 2m temperature ($^o$C) in Bodélé and Cape Verde.*

Results are presented in Table 2 for the surface meteorological variables, 10m wind speed and the 2m temperature, and in Table 3 for the total precipitation and the surface mineral dust concentrations. For each variable, the presented values are the mean value (averaged over the whole month of September 2021), the bias and the correlation. The line (D-1) is always empty since the bias and the correlation of itself compared to (D-1) gives the values of 0 and 1, respectively.

For the 10m wind speed, it is noticeable that the mean value does not evolve a lot during the forecast. In Bodélé, the bias is always lower than -0.2 m.s$^{-1}$, the negative value meaning that the (D-1) analysis simulation was the one with the highest wind speed. In Cape Verde, the mean value is larger, but the bias lower. With a maximum of -0.11, and negative or positive values, there is not a lot of variability for this site. An important point is the correlation of the leads compared to the (D-1) one: the values are very low, for the two sites: between -0.27 and 0.27 for the maximum values. It means that, from one day to the next one, the time-series are varying a lot in frequency. If the mean values are close, the maximum of wind speed are not at the same time.

For the 2m temperature, we can also observe an important lack of variability for the mean values. The 2m temperature is higher ($\approx$ 32 $^o$C in average) in Bodélé than in Cape Verde ($\approx$ 24 $^o$C in average), clearly showing the difference between a desert and a maritim-infuenced air around an Island. The bias is negative over Bodélé showing that the forecast tends to underestimate the temperature compared to the lead (D-1). The bias is positive over Cape Verde but the values are so low that it is negligible. Over Bodélé, the correlation remains high (between 0.75 and 0.9) showing that the forecast over the desert is very stable. It is not the case over Cape Verde, with a large variability, between 0.08 and 0.57. And the decrease of the correlation is not linear

with the increasing lead: (D+4) has a correlation of 0.29 when (D+2) has a correlation of 0.08. The fact to have a stable forecast over land does not mean that the forecast is stable over the Atlantic sea, the meteorological systems being completely different.

| | Total precipitation (kg.m$^{-2}$.h$^{-1}$ × 100) | | | Mineral dust conc. ($\mu$g.m$^{-3}$) | | |
|---|---|---|---|---|---|---|
| Lead | Mean | Bias | R | Mean | Bias | R |
| *Bodélé* | | | | | | |
| (D-1) | 1.14 | | | 1291.0 | | |
| (D+0) | 1.12 | -0.021 | 0.02 | 1248.9 | -42.111 | 0.17 |
| (D+1) | 0.70 | -0.443 | -0.02 | 1125.9 | -165.070 | -0.09 |
| (D+2) | 0.86 | -0.283 | -0.04 | 1290.4 | -0.584 | -0.10 |
| (D+3) | 0.28 | -0.861 | -0.04 | 1352.2 | 61.142 | -0.22 |
| (D+4) | 1.92 | 0.776 | -0.06 | 1206.6 | -84.424 | -0.13 |
| *Cape Verde* | | | | | | |
| (D-1) | 2.81 | | | 195.6 | | |
| (D+0) | 2.51 | -0.305 | 0.00 | 200.3 | 4.703 | 0.13 |
| (D+1) | 3.12 | 0.304 | -0.07 | 197.9 | 2.336 | -0.30 |
| (D+2) | 2.53 | -0.286 | -0.06 | 193.1 | -2.446 | 0.14 |
| (D+3) | 6.43 | 3.612 | -0.06 | 185.5 | -10.045 | 0.19 |
| (D+4) | 3.25 | 0.434 | -0.07 | 173.5 | -22.108 | -0.11 |

**Table 3.** *Statistical scores for the daily cumulated total precipitation (kg.m$^{-2}$.h$^{-1}$ × 100) and surface mineral dust concentrations ($\mu$g.m$^{-3}$) in Bodélé and Cape Verde.*

The same type of scores is displayed in Table 3 for the daily cumulated total precipitation (kg.m$^{-2}$.h$^{-1}$ × 100) and the surface mineral dust concentrations ($\mu$g.m$^{-3}$) in Bodélé and Cape Verde. For the precipitation, it is remarkable to see that the correlation is always close to zero. It means that, from day to day, the precipitation varies a lot for a specific location. It is logical since precipitation is a threshold process, not continuous in space and time, contrarily to the temperature or the wind. The bias is important, both in Bodélé and Cape Verde: it corresponds to the fact to have a forecast with a precipitation and the next lead without for the same place and time. For this process, the statistical scores show that the forecast is very variable from one day to another.

For the mineral dust concentrations, the correlation is also low for both sites. Over Bodélé, the values are between -0.22 for (D+3) and +0.17 for (D+0), and over Cape Verde, values are between -0.30 for (D+1) and +0.19 for (D+3). The bias is non negligible and may reach 10% of the mean values. As for the other parameters, there is no a regular decrease with an increasing lead: the system is chaotic and the unstability of the forecat for dust concentrations is the reflect of the unstability of the mean wind speed over sources areas, then emissions, then transport, then concentrations at remote locations. A common result to all parameters is that the best scores (for bias and correlation) are often obtained for lead time close to the analysis (D-1).

## 4 Merging the forecast leads to make an ensemble

The previous results showed that small variations of meteorological variables may change a lot mineral dust concentrations after long-range transport. This quantification was made with only model results in order to quantify the model's variability.

For some locations, it is however possible to compare the AOD to AERONET measurements. During the studied period, four stations are present in the modelled domain and have available data: Zinder (Niger), Banizoumbou (Niger), Cinzana (Mali) and Cape Verde. Note that, unfortunately, there is no measurement for this period at Bodélé. Using these data, it is possible to calculate statistical scores between the modelled forecast and the measurements.

An added value in this study, it that it is also possible to add two model realizations. Considering that the various forecast leads are performed each time with a new meteorology, then natural emissions (here mineral dust emissions), we can consider all these leads as independant simulations. They are thus similar to ensemble forecast members, usually made the same day but with perturbed initial conditions. As presented in Figure 2, for one date we have six simulations. It is possible to make the hypothesis that these six forecast leads are equivalent to six ensemble members. To test this approach, we use the time-series at the four locations where AERONET measurements are available to create a two new sets called ENSmean and ENS median:

- ENSmean corresponds to the mean averaged value of the six members,

- ENSmedian corresponds to the median of the members. Having only size members, this values is in fact the mean average of the $3^{th}$ and $4^{th}$ members.

### 4.1 Scores during the CADDIWA period

Statistical scores are first calculated for the period of the CADDIWA experiment, for the period September 1 to 31, 2021. Results are presented in Table 4. For each site and each parameter (correlation $R$, RMSE or bias), the best score is bolded. It shows that model realizations are close from each other but remain different to the measurements. The variability of the forecast is lower than the difference between measurements and models. It means that the model systematically underestimates the AOD whatever the perturbations included in each forcast realization. The bias is negative for all sites over land (Zinder, Banizoumbou and Cinzana) and over sea, the island of Cape Verde. The bias is smaller over this latter site.

For the correlation, the best value is obtained for the ENSmean lead for two sites out of four: Cape Verde and Zinder. For Cinzana and Banizoumbou the best correlation is obtained for (D+3) and (D+2), respectively. It means that (i) the best scores is not for the "analysis" lead (D-1) as it could be expected, (ii) the combination of lead leading to ENS may be the best forecast. For the bias, results are different: the lower biases are not for ENS. In Cape Verde, the lower bias is for (D-1). For the other sites, they are for (D+3) and (D+4) forecasts. Note that even if ENS has not the best score, it is no the worse too. These scores show that the best forecast lead is not always the closest to the analysis. It also shows that the use of a "ensemble" lead may provide good results.

For the RMSE, some best scores are also obtained for the ENSmean configuration, showing that the merge of the lead may reduce the model error. For the bias, the values remain very close from one lead to antoher and there is no really a best configuration.

| | Aerosol Optical Depth | | | | |
| SITE | $\overline{obs}$ | $\overline{model}$ | R | RMSE | bias |
|---|---|---|---|---|---|
| *(D-1)* | | | | | |
| CapeVerde | 0.41 | 0.33 | 0.75 | 0.16 | **-0.07** |
| Cinzana | 0.28 | 0.11 | 0.44 | 0.23 | -0.18 |
| Banizoumbou | 0.43 | 0.13 | **0.76** | 0.37 | -0.30 |
| Zinder | 0.55 | 0.26 | 0.54 | 0.54 | -0.29 |
| *(D+0)* | | | | | |
| CapeVerde | 0.41 | 0.33 | 0.75 | 0.16 | **-0.07** |
| Cinzana | 0.28 | 0.11 | 0.45 | 0.22 | -0.18 |
| Banizoumbou | 0.43 | 0.13 | **0.76** | 0.37 | -0.30 |
| Zinder | 0.55 | 0.26 | 0.55 | 0.51 | 0.40 |
| *(D+1)* | | | | | |
| CapeVerde | 0.41 | 0.33 | 0.76 | 0.16 | -0.08 |
| Cinzana | 0.28 | 0.10 | 0.43 | 0.23 | -0.18 |
| Banizoumbou | 0.43 | 0.14 | 0.74 | 0.37 | -0.30 |
| Zinder | 0.55 | 0.26 | 0.46 | 0.41 | -0.29 |
| *(D+2)* | | | | | |
| CapeVerde | 0.41 | 0.32 | 0.76 | 0.16 | -0.08 |
| Cinzana | 0.28 | 0.11 | 0.36 | 0.23 | -0.18 |
| Banizoumbou | 0.43 | 0.14 | **0.76** | 0.37 | -0.30 |
| Zinder | 0.55 | 0.27 | 0.52 | **0.40** | -0.28 |
| *(D+3)* | | | | | |
| CapeVerde | 0.41 | 0.30 | **0.78** | 0.17 | -0.11 |
| Cinzana | 0.28 | 0.12 | **0.48** | **0.21** | -0.16 |
| Banizoumbou | 0.43 | 0.14 | 0.68 | **0.36** | -0.29 |
| Zinder | 0.55 | 0.27 | 0.49 | **0.40** | **-0.27** |
| *(D+4)* | | | | | |
| CapeVerde | 0.41 | 0.32 | 0.67 | 0.21 | -0.09 |
| Cinzana | 0.28 | 0.13 | 0.35 | **0.21** | **-0.15** |
| Banizoumbou | 0.43 | 0.17 | 0.39 | 0.37 | **-0.27** |
| Zinder | 0.55 | 0.23 | 0.56 | 0.43 | -0.32 |
| *Mean* | | | | | |
| CapeVerde | 0.41 | 0.32 | **0.78** | **0.15** | -0.08 |
| Cinzana | 0.28 | 0.11 | 0.45 | 0.22 | -0.17 |
| Banizoumbou | 0.43 | 0.14 | 0.75 | 0.37 | -0.29 |
| Zinder | 0.55 | 0.26 | **0.59** | **0.40** | -0.29 |
| *Median* | | | | | |
| CapeVerde | 0.41 | 0.33 | 0.76 | 0.16 | -0.08 |
| Cinzana | 0.28 | 0.11 | 0.43 | 0.23 | -0.18 |
| Banizoumbou | 0.43 | 0.14 | 0.75 | 0.37 | -0.30 |
| Zinder | 0.55 | 0.26 | 0.55 | 0.41 | -0.29 |

**Table 4.** *For the period of September 1 to 31, 2021, correlation (R), RMSE and bias calculated between the AERONET Aerosol Optical Depth measurements and the modelled results. Results are presented for four sites, Cape Verde, Cinzana, Banizoumbou and Zinder and for six forecast leads, from (D-1) to (D+4). Two additional forecast leads called ENSmean and ENSmedian represent the mean average of the previous six leads and the median, respectively. The best scores for each sites and among all leads are bolded.*

Table 5 summarizes the results presented in Table 4 by recalculating the scores but for the four stations (Cape Verde, Cinzana, Banizoumbou and Zinder) together. As for the previous results. the best scores are bolded as a function of the forecast lead.

| Aerosol Optical Depth | | | | |
| --- | --- | --- | --- | --- |
| Lead | $R_s$ | R | RMSE | bias |
| (D-1) | 0.53 | 0.62 | 0.29 | -0.21 |
| (D+0) | 0.53 | 0.63 | 0.29 | -0.21 |
| (D+1) | 0.57 | 0.60 | 0.29 | -0.21 |
| (D+2) | 0.60 | 0.60 | 0.29 | -0.21 |
| (D+3) | **0.62** | 0.61 | 0.29 | -0.21 |
| (D+4) | 0.45 | 0.49 | 0.29 | -0.21 |
| ENSmean | 0.57 | **0.64** | 0.29 | -0.21 |
| ENSmedian | 0.54 | 0.62 | 0.29 | -0.21 |

**Table 5.** *For the period of September 1 to 31, 2021, spatial and temporal correlation, RMSE and bias for each lead and as average for the four stations, for the AOD measured by AERONET and modelled by WRF-CHIMERE.*

The spatial correlation $R_s$ is the best for (D+3), the correlation $R$ is the best for the ENSmean forecast. The RMSE and bias remain the same for all model realizations.

| 2m temperature ($^o$C) | | | | |
| --- | --- | --- | --- | --- |
| Lead | $R_s$ | R | RMSE | bias |
| (D-1) | 0.91 | **0.34** | 0.07 | 0.33 |
| (D+0) | 0.91 | **0.34** | 0.07 | 0.33 |
| (D+1) | 0.91 | 0.33 | 0.07 | 0.29 |
| (D+2) | **0.92** | 0.31 | 0.07 | 0.18 |
| (D+3) | **0.92** | 0.24 | 0.08 | 0.24 |
| (D+4) | 0.89 | 0.21 | 0.07 | **0.15** |
| ENSmean | 0.91 | 0.32 | 0.07 | 0.25 |
| ENSmedian | 0.91 | 0.33 | 0.07 | 0.30 |

**Table 6.** *For the period of September 1 to 31, 2021, spatial and temporal correlation, RMSE and bias for each lead and as average for the four stations, for the 2m temperature measurements provided by UWYO and modelled by WRF-CHIMERE.*

The same type of scores is presented for the 2m temperature (Table 6) and the 10m wind speed (Table 7). Scores are calculated using the UWYO meteorological data. They are hourly but the problem is that they are recorded in integer, decreasing their accuracy and possibly biasing the calculation of differences between observations and model results. It is interesting to explore the statistical scores of these two parameters since they are good proxys of mineral dust emissions: the 10m wind speed is directly used for the saltation process, via the friction velocity $u_*$, and the 2m temperature is used to diagnosed the additional free convection velocity $w_*$, (Menut et al., 2013).

For the 2m temperature, the bias is positive as confirmed by the time-series presented in Figure 5. This bias is varying a lot between leads and the lower bias is for (D+4). The spatial correlation, $R_s$, is high but more or less constant between leads with values from 0.89 to 0.92. It means that the differences between sites remain close between the leads. The temporal correlation $R$ ranges from 0.24 to 0.34 for (D-1). The ensemble leads provide correct scores with $R$=0.32 and 0.33.

For the 10m wind speed, results are more variable. The spatial correlation ranges from 0.79 to 0.89 for (D+3). There is no regular decrease of the score with the lead. The ensemble is not the best score but with $R_s$=0.84, the spatial correlation is better

than (D-1) or (D+0). The temporal correlation is not correct and close to 0. As presented in Figure 5, the modelled wind speed does not follow the day to day variations observed with the measurements. But to compare observed and modelled wind speed remain challenging: first for the integers recorded with the stations and second with the differences of representativity with a specific observations site and, on the other hand, a model cell of a few tens of squared kilometers. Finally, the scores for 2m
temperature and 10m wind speed are not able to explain completely the scores obtained with the ensemble lead.

| 10m wind speed (m.s$^{-1}$) | | | | |
|---|---|---|---|---|
| Lead | $R_s$ | R | RMSE | bias |
| (D-1) | 0.81 | 0.05 | 0.41 | **0.02** |
| (D+0) | 0.79 | 0.04 | 0.41 | **0.02** |
| (D+1) | 0.82 | -0.00 | 0.43 | 0.11 |
| (D+2) | 0.87 | 0.05 | 0.42 | 0.15 |
| (D+3) | **0.89** | 0.02 | **0.40** | 0.13 |
| (D+4) | 0.82 | **0.12** | 0.42 | 0.13 |
| ENSmean | 0.84 | 0.04 | **0.40** | 0.10 |
| ENSmedian | 0.84 | 0.05 | **0.40** | 0.07 |

**Table 7.** *For the period of September 1 to 31, 2021, spatial and temporal correlation, RMSE and bias for each lead and as average for the four stations, for the 10m wind speed measurements provided by UWYO and modelled by WRF-CHIMERE.*

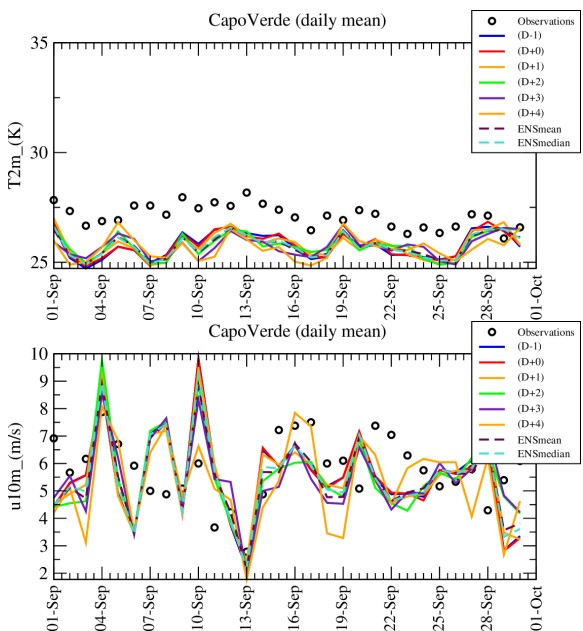

**Figure 5.** *Time-series of 2m temperature (°C) and 10m wind speed (m.s$^{-1}$) measured (UWYO database) and modelled during the forecast and for several leads. The last time-series, called ENSmean and ENSmedian, are the mean averaged and the median values of the previous leads, from (D-1) to (D+4).*

## 4.2 Scores during the extended forecast period

In order to have more statistically robust results, the complete modelled period is now presented: for the period August 15 to November 1, 2021. This period is around the CADDIWA experiment and coresponds to the period when the forecast system was running, i.e. 2.5 months. Results are presented as time-series in Figure 6 for the daily averaged AOD.

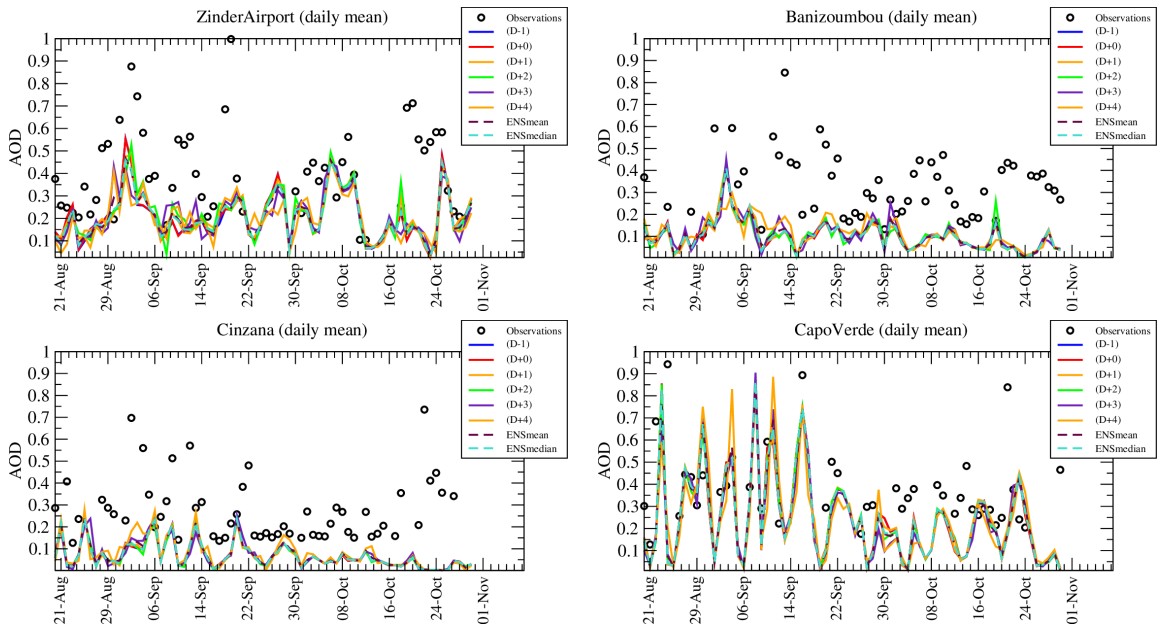

**Figure 6.** *Time-series of Aerosol Optical Depth (AOD) measured by AERONET and modelled during the forecast and for several leads. The last time-series, called ENSmean and ENSmedian, are the mean averaged and the median values of the previous leads, from (D-1) to (D+4).*

It is noticeable that the month of September (compared to August and October) is not the month with the highest AOD: values are of the same order of magnitude over the whole period. Except in Cape Verde where the largest peaks are observed during September 2021, corresponding to the CADDIWA measurements campaign.

Statistical scores are presented in Table 8 in the same way that in Table 4 but this time for a longer period. Over this period, the availability of hourly measurements is 62.5%, 77.8%, 79.2% and 77.8% for Cape Verde, Cinzana, Banizoumbou and Zinder stations, respectively. For this longer period, the best correlation are not for the ensemble leads, except for the RMSE in Cape Verde and Zinder and for the bias in Cape Verde. For the correlation, the best scores are now for the first forecast leads, i.e (D-1) and (D+0). All in all, the scores are very close from one lead to the next one.

Table 9 summarizes the results presented in Table 8 as in Table 5. The best spatial correlation is again for (D+3) when the best temporal correlation is obtained for the leads (D-1) and (D+0). For the RMSE, scores are very close between leads, and the best values are for (D-1), (D+0), (D+1) but also for the ensemble, both with ENSmean and ENSmedian.

| SITE | Aerosol Optical Depth | | | | |
| | $\overline{obs}$ | $\overline{model}$ | R | RMSE | bias |
| --- | --- | --- | --- | --- | --- |
| *(D-1)* | | | | | |
| CapeVerde | 0.41 | 0.28 | 0.48 | **0.26** | **-0.13** |
| Cinzana | 0.29 | 0.07 | **0.41** | 0.27 | -0.22 |
| Banizoumbou | 0.36 | 0.11 | **0.60** | 0.30 | -0.25 |
| Zinder | 0.44 | 0.22 | **0.42** | 0.32 | -0.22 |
| *(D+0)* | | | | | |
| CapeVerde | 0.41 | 0.28 | 0.48 | **0.26** | **-0.13** |
| Cinzana | 0.29 | 0.07 | **0.41** | 0.27 | -0.22 |
| Banizoumbou | 0.36 | 0.11 | **0.60** | 0.30 | -0.25 |
| Zinder | 0.44 | 0.22 | **0.42** | 0.32 | -0.22 |
| *(D+1)* | | | | | |
| CapeVerde | 0.41 | 0.28 | 0.48 | **0.26** | **-0.13** |
| Cinzana | 0.29 | 0.07 | 0.41 | 0.27 | -0.22 |
| Banizoumbou | 0.36 | 0.11 | **0.60** | 0.30 | -0.25 |
| Zinder | 0.44 | 0.22 | 0.37 | **0.32** | -0.22 |
| *(D+2)* | | | | | |
| CapeVerde | 0.41 | 0.27 | **0.49** | **0.26** | **-0.13** |
| Cinzana | 0.29 | 0.07 | 0.36 | 0.27 | -0.21 |
| Banizoumbou | 0.36 | 0.11 | 0.56 | 0.30 | -0.25 |
| Zinder | 0.44 | 0.23 | 0.40 | 0.32 | -0.22 |
| *(D+3)* | | | | | |
| CapeVerde | 0.41 | 0.26 | 0.42 | 0.27 | -0.15 |
| Cinzana | 0.29 | 0.08 | 0.34 | **0.27** | -0.21 |
| Banizoumbou | 0.36 | 0.12 | 0.50 | 0.30 | **-0.24** |
| Zinder | 0.44 | 0.23 | 0.31 | 0.32 | **-0.21** |
| *(D+4)* | | | | | |
| CapeVerde | 0.41 | 0.26 | 0.41 | 0.27 | -0.15 |
| Cinzana | 0.29 | 0.09 | 0.24 | **0.27** | **-0.20** |
| Banizoumbou | 0.36 | 0.12 | 0.36 | **0.30** | -0.24 |
| Zinder | 0.44 | 0.21 | 0.30 | 0.34 | -0.23 |
| *Mean* | | | | | |
| CapeVerde | 0.41 | 0.27 | 0.48 | **0.26** | -0.14 |
| Cinzana | 0.29 | 0.08 | 0.38 | 0.27 | -0.21 |
| Banizoumbou | 0.36 | 0.11 | 0.58 | **0.30** | -0.25 |
| Zinder | 0.44 | 0.22 | 0.41 | 0.32 | -0.22 |
| *Median* | | | | | |
| CapeVerde | 0.41 | 0.28 | 0.48 | **0.26** | **-0.13** |
| Cinzana | 0.29 | 0.07 | 0.39 | 0.27 | -0.22 |
| Banizoumbou | 0.36 | 0.11 | 0.59 | 0.30 | -0.25 |
| Zinder | 0.44 | 0.22 | 0.39 | **0.32** | -0.22 |

**Table 8.** *For the period August 15 to November 1, 2021, correlation (R), RMSE and bias calculated between the AERONET Aerosol Optical Depth measurements and the modelled results. Results are presented for four sites, Cape Verde, Cinzana, Banizoumbou and Zinder and for six forecast leads, from (D-1) to (D+4). Two additional forecast leads called ENSmean and ENSmedian represent the mean average of the previous six leads and the median, respectively. The best scores for each sites and among all leads are bolded.*

| | Aerosol Optical Depth | | | |
|---|---|---|---|---|
| Lead | $R_s$ | R | RMSE | bias |
| (D-1) | 0.87 | **0.48** | 0.29 | -0.20 |
| (D+0) | 0.87 | **0.48** | 0.29 | -0.20 |
| (D+1) | 0.87 | 0.46 | 0.29 | -0.20 |
| (D+2) | 0.90 | 0.45 | 0.29 | -0.20 |
| (D+3) | **0.91** | 0.39 | 0.29 | -0.20 |
| (D+4) | 0.86 | 0.33 | 0.30 | -0.20 |
| ENSmean | 0.87 | 0.46 | 0.29 | -0.20 |
| ENSmedian | 0.87 | 0.46 | 0.29 | -0.20 |

**Table 9.** *Comparison between observations and model for the AOD. For the period August 15 to November 1, 2021, spatial and temporal correlation, RMSE and bias for each lead and as average for the four stations.*

## 5   Conclusions

In this study, the first goal was to examine the variability of the forecast as a function of the lead time and for each forecasted day. This forecast was daily performed for six days, during the period August to October 2021 and as support for the CAD-DIWA field campaign. For meteorological variables (2m temperature, 10m wind speed, total precipitation rate) and surface concentrations of mineral dust, the day to day variability was quantified. The performances of the forecast over two sites were performed, Bodélé (desert area and important source of dust) and Cape Verde (where the measurements of DACCIWA were coordinated). It has been shown that the wind speed is highly variable for day to day forecast while the temperature is stable over land but more variable over sea and shores (Cape Verde being a group of little Islands). The less stable parameter is the precipitation at one location when the model may forecast an event one day and not at all the day after.

First, one goal of the study was to examine if a large forecast variability at one site (such as Bodélé) may have a visible impact at a downwind remote site (such as Cape Verde). No evidence of a transport of variability (or a transport of stability) was found during the forecast. The large variability of wind speed, precipitation and temperature induce a large variability of the surface concentration of mineral dust. Between forecast leads, large differences were found both for the correlation and the bias. Considering the model configuration used for this study where no direct and indirect effects of aerosols on meteorology and only mineral dust as natural emissions was taken into account, this variability could be underestimated. A next study could be to replay this forecast with a model version including all anthropogenic and natural emissions in the CHIMERE model with an exhaustive evaluation with the measurements of the experiment to come.

Second, a new way combining forecast leads was tested to improve the predictions. Considering that several forecast leads may be considered as the members of an ensemble, they are combined from (D-1) to (D+4) for all coinciding dates computing the mean and median values. These new "forecast leads" are compared, with all others members, to the Aerosol Optical Depth measurements of AERONET using correlation, RMSE and bias statistics. It is noticeable that the forecast is not impaired when increasing the lead time. But it is also noticeable that out of four sites, the best scores for two sites are with the ensemble for the period of the CADDIWA campaign. It is not the case for an extended analyzed period, highlighting that the scores are close from one lead to another. The ensemble methodology provides the best scores when the AOD values are the most

important and the most variables in time. This result opens perspectives for forecasting in general. It would be interesting to test this hypothesis on operational systems: if the combination of the previous forecasts allows to improve the initial conditions of a new forecast, it would allow to perform less ensemble simulations for the same day and thus to reduce considerably the computing cost.

## Appendix A:  Complementary analysis with the meteorological variables

### A1    Time-series of meteorological variables

In addition to the time-series presented in Section 3.1, the same results are here presented for 2m temperature, 10m wind speed and precipitation rate.

Figure A1 presents time-series and differences for 2m temperature ($^oC$) in Bodélé and Cape Verde. The 11 and 18 September are noted on the Figure with a black vertical line. In Bodélé the temperature is higher than in Cape Verde, with values between
30 and 35 $^oC$. During the month of September the temperature decreases regularly. The days of 11 and 18 Septembre correspond to periods with the highest temperature values. In Cape Verde, there is no similar trend: the daily averaged temperature remains around 25 $^oC$ showing the maritime characteristic of the Cape Verde environment. The differences are low and oscillate between -2 and +2 $^oC$. The longer the forecast, the greater the variability. In Cape Verde, the variability is lower and between -1 and +1 $^oC$. As in Bodélé, the largest differences with (D-1) are obtained with (D+4). The forecast of temperature appear to be
relatively stable, the differences logically growing with the increasing leads.

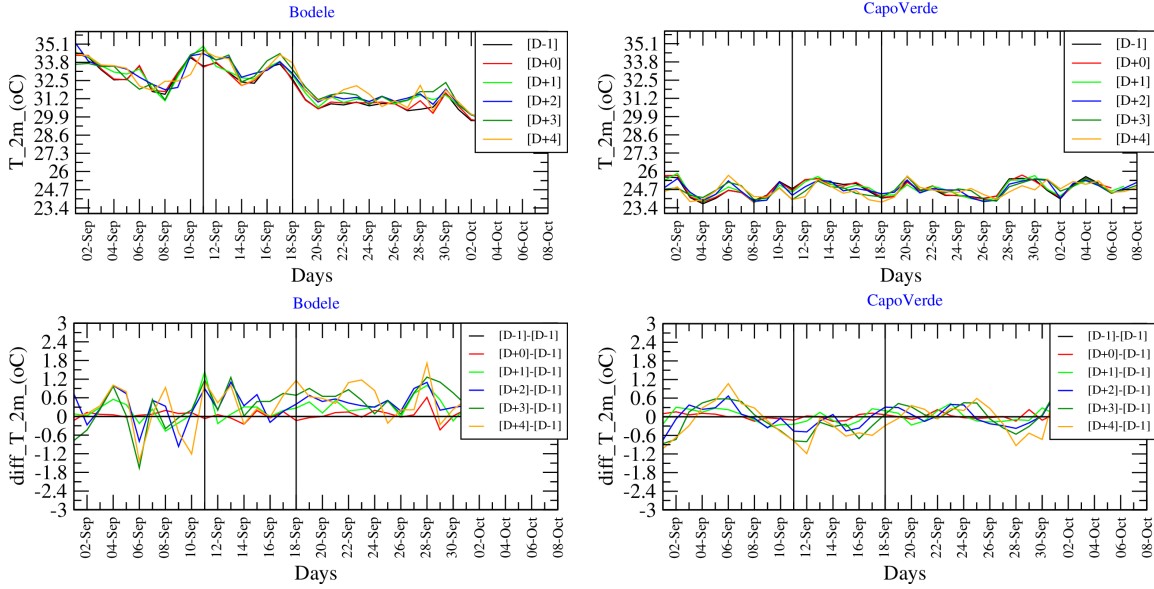

**Figure A1.** *Time-series of 2m temperature ($^oC$) for each lead and of differences between leads.*

Figure A2 presents results for 10m wind speed. Values are lower in Bodélé (middle of desert) than in Cape Verde (an island). In Bodélé, daily averaged values are between 2 and 7 m.s$^{-1}$, values lower than minimum value generally required for dust erosion over barren soils. But hourly values may be larger and the model uses a Weibull distribution to take into account the sub-hour and the sub-grid spatial variability, (Menut, 2018). It is noticeable the two days of 11 and 18 September don't correspond to high wind speed value, as well as the days before. In Cape Verde, the wind speed values are between 3 and 10 m.s$^{-1}$, with day to day variability higher than in Bodélé. Some days shows high values such as 5 and 11 September. It is the signature, close to the surface, to large scale meteorological motions.

On Figure A2, differences are also presented. Differences are of the same order of magnitude between the two locations. For (D+0)-(D-1), differences are of maximum $\pm$ 0.5 m.s$^{-1}$, when higher values are calculated for (D+4)-(D-1) with maximum around $\pm$ 3 m.s$^{-1}$. It is noticeable that the differences increase with the lead: the more distant the forecast, the greater the difference between the leads. For these differences, there is no systematic bias: they can be negative or positive, showing a variability not due to large scale and/or persistent atmospheric systems, but much more regional variability, with an higher temporal frequency. More specifically for the 11 September, when the absolute values shows a peak in Cape Verde, the differences show this peak was predicted late: four days before, (D+4) forecast, the peak is 6 m.s$^{-1}$, when it is 9 m.s$^{-1}$ for (D+0). The difference is then -3 m.s$^{-1}$, one of the most important during the whole modelled period.

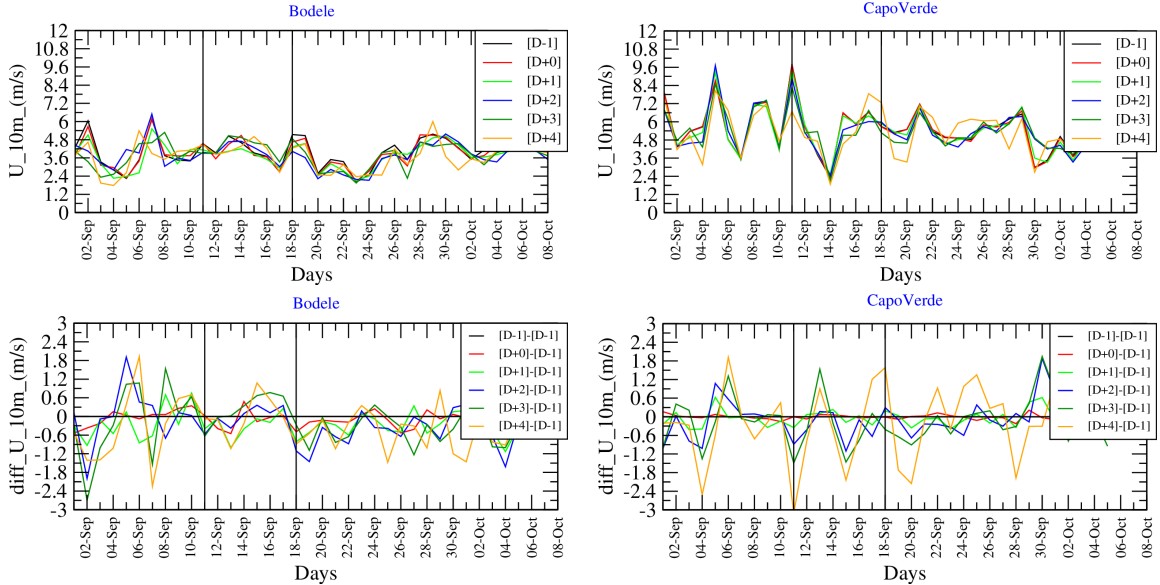

**Figure A2.** *Time-series of 10m wind speed (m.s$^{-1}$) for each lead and of differences between leads.*

Figure A3 presents the same kind of time-series but for total precipitation in kg.m$^{-2}$.h$^{-1}$ $\times$100. The time series show that only a few periods had precipitation episodes for the two sites of Bodélé and Cape Verde. In Bodélé, the two periods with rain are the 6 and 15 September. In Cape Verde, three episodes are modelled, 6, 14 September and 5 October (the last one being out

of the current analyzed period). For the first episode in Bodélé, 6 September, it appears only for the forecast lead (D+4). For the other forecasts, closer in time, there is no precipitation. For the second episode, a time variability is observed: depending on the lead, the precipitation have similar magnitude but is forecasted on 14, 15 or 16 September. The difference show the forecast is mostly overestimated compared to the analysis of (D-1). In Cape Verde, the several precipitation epiosed are also varying in time. If the first episode is over-estimated for the (D+4) lead, it is finally underestimated by the other leads, from (D+1) to (D+3). The second episode is forecasted with less variability in time, all forecasts being for the 13 or 14 September only. The maginitudes are close between leads, with only a low underestimation compared to (D-1). Finally, the forecast is less variable in Cape Verde than in Bodélé.

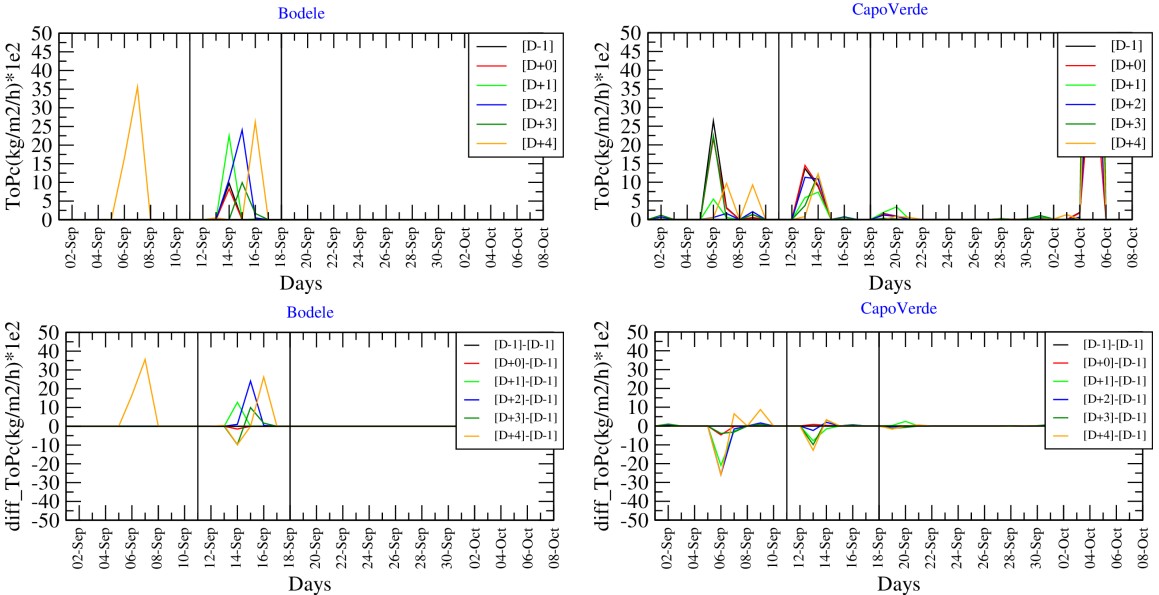

**Figure A3.** *Time-series of total precipitation (kg.m$^{-2}$.h$^{-1}$ ×100) for each lead and of differences between leads.*

## A2 Maps of wind speed and precipitation

Results are presented as maps for one date, the 11 September at 12:00 UTC. Figure A4 first presents maps for the 10-m wind-speed (m.s$^{-1}$) and total precipitation (kg.m$^{-2}$.h$^{-1}$). On the left panel, the absolute value of the forecast lead (D-1) is presented. On the right panel, the differences between the leads (D-4) and (D-1). Note that for the wind speed, the wind vectors are superimposed. For the 10 m wind speed, the values of (D-1) show moderate values (between 0 and 3 m.s$^{-1}$), except over Mauritania with maximum values $\approx$ 15 m.s$^{-1}$. The wind speed is larger over the Atlantic sea with values $\approx$ 8 m.s$^{-1}$ near Cape Verde. The Cape Verde site is inbetween two different air masses: one coming from the South and evolving along the African coast, the second one, on the west side of Cape Verde and coming from the North. It results of low wind locally in Cape Verde. The map of differences show the same pattern, meaning that this structure changes during the forecast: the (D+4)

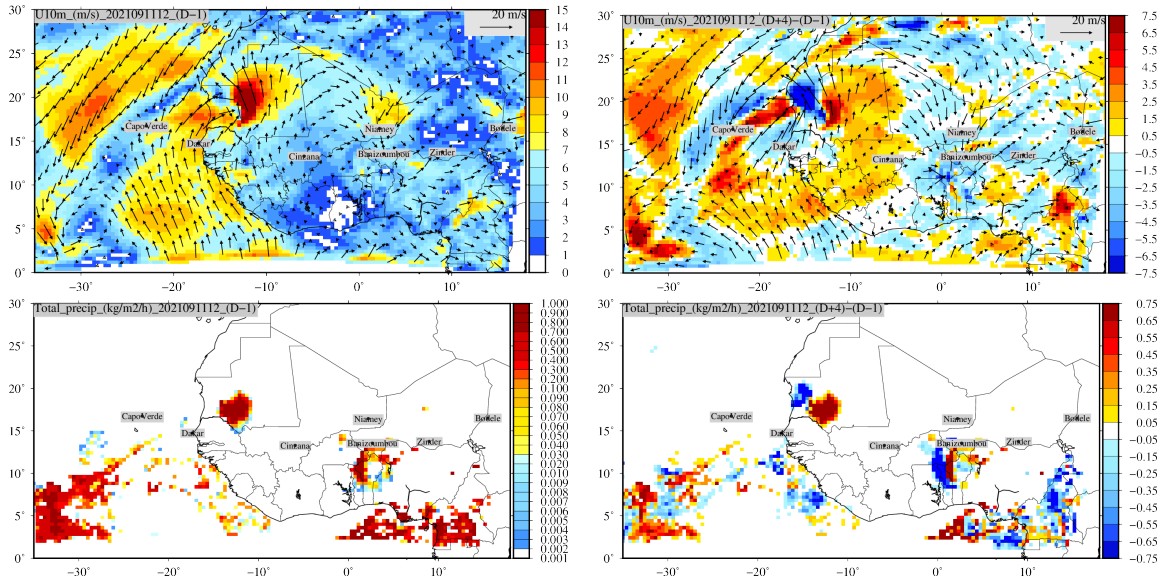

**Figure A4.** *Maps of 10-m wind-speed ($m.s^{-1}$) and total precipitation ($kg.m^{-2}.h^{-1}$) for the 11 September at 12:00 UTC. Values are displayed (left) for the forecast lead (D-1) and (right) for the differences between the forecast leads (D-4)-(D-1).*

forecast shows negative values, meaning that the wind speed is higher for (D-1) than (D+4). It means that the strong gradient, from north-east to south-west and flewing over Cape Verde, observed for (D-1), was not present for the forecast four days in advance.

For the total precipitation, Figure A4, the results on the map show very localized events. Over Africa and over Mauritania, the large amount of precipitations ($\approx 1$ $kg.m^{-2}.h^{-1}$) is colocated with the large 10 m wind speed values.Other precipitation events are modelled more in the South, both over the Atlantic Sea and the Gulf of Guinea, for a latitude below $10^oN$. The differences map shows negative and postive values: it is the mark of a change in wind speed and direction, then location of the precipitation. But whatever the location, each time a precipitation was forecasted, each time it occured, even if this is not strictly at the same place. For this day, no precipitation was forecasted over Cape Verde and this forecast remained stable during the several forecast leads.

## A3 Vertical cross-sections of mineral dust concentrations and rain

Figure A5 presents vertical cross-section of mineral dust concentrations ($\mu g.m^{-3}$) and precipitation rate (in $kg.kg^{-1}.10^6$) at isolatitude $17^oN$ for the forecast lead of (D-1) and the difference between (D+4) and (D-1). The same day and hour, than for the horizontal maps, are selected for these results.

The goal of these Figures is to present the vertical extent of the possible differences between the leads. For the mineral dust concentrations, the large surface concentrations extend vertically until 3000 m. And concentrations are non negligible until 7000 m.At the longitude of Cape Verde, -23 $^oW$, dust concentrations are large, but the forecast is very variable. The differences

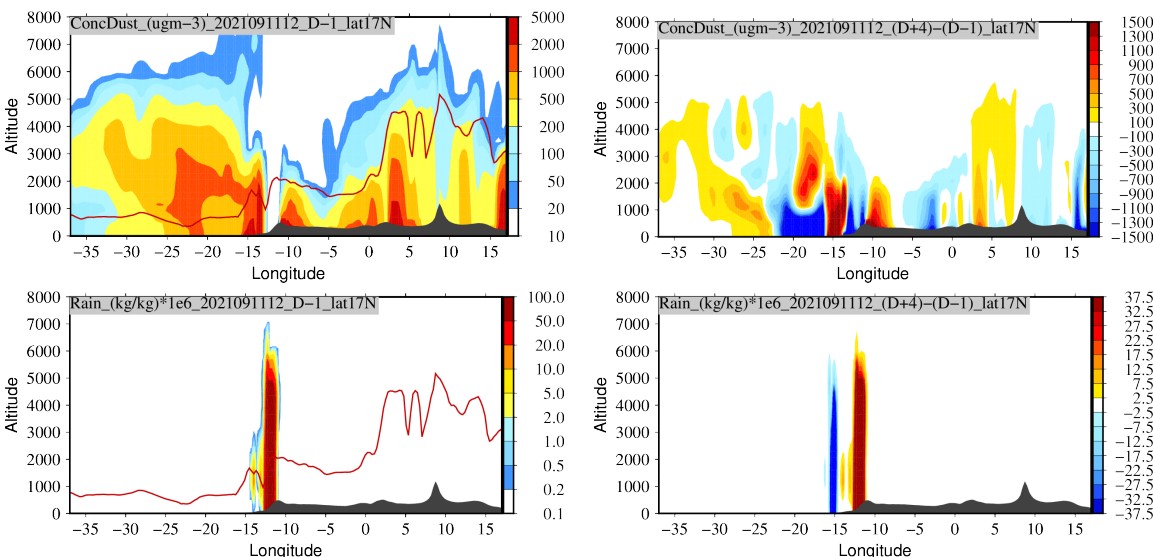

**Figure A5.** *Vertical cross-section of mineral dust concentrations ($\mu g\ m^{-3}$) and precipitation rate (in $kg.kg^{-1}.10^6$) at isolatitude 17$^o$N. Figures represent the same date, 11 September 2021 at 12:00 UTC. [left] absolute values for forecast lead (D-1) and [right] differences between forecast leads (D+4)-(D-1). The line in red is the boundary layer height.*

shows maximum values between -20 and -10 $^o$W. Around -20 $^o$W, the vertical structure shows negative values close to the surface but positive values between 1500 and 3000 m, above the boundary layer. It means that the wind direction changed

between the forecast leads but also the vertical distribution of the dust plume coming from Africa. It explains the differences for the surface concentrations and should also have an impact on AOD (see section 4).

The vertical profile of rain shows for this day a large event at longitude ≈ -13 $^o$W. It corresponds to the event seen on Figure A4 over the south-west of Mauritania. The vertical cross-section of differences shows negative then positive values: the wind being faster as the forecast is close from the current day, the precipitation is transported faster and then appears as

positive for longitude -13 $^o$W and negative in longitude -16 $^o$W. If the horizontal transport is changing with leads, the vertical structure remains the same with a maximum at 6000 m.

*Code availability.* The CHIMERE v2020 model is available on its dedicated web site https://www.lmd.polytechnique.fr and for download at https://doi.org/10.14768/8afd9058-909c-4827-94b8-69f05f7bb46d.

*Data availability.* All data used in this study, as well as the data required to run the simulations, are available on the CHIMERE web site

download page https://doi.org/10.14768/8afd9058-909c-4827-94b8-69f05f7bb46d.

*Author contributions.* The author made completely the study.

*Competing interests.* The author declares that he has no conflict of interest.

*Acknowledgements.* The authors thank the OASIS modeling team for their support with the OASIS coupler, the WRF developers team for the free use of their model. We thank the investigators and staff who maintain and provide the AERONET data (https://aeronet.gsfc.nasa.gov/).

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
