# Peer review of "Variability and combination as ensemble of mineral dust forecast during the 2021 CADDIWA experiment, using the WRF 3.7.1 and CHIMERE v2020r3 models."

_Geoscientific Model Development, 2022_

## Referee Comment (RC1)

This paper presents an evaluation of a forecast of mineral dust in support to the CADDIWA experiment over the western part of Africa. Also, an innovative way to produce an ensemble forecast is presented by using lead time of a single model instead of using several models. The coupled regional model WRF-CHIMERE is deployed in forecast mode during the summer 2021. I think this publication is suitable for publication after the following corrections.

**Major comments**

- My feeling is that the new methodology (using an ensemble of forecast lead) proposed in this paper could be applied to the general purpose of weather forecast. Since the WRF-CHIMERE also deliver data, I strongly suggest to apply the method to the main meteorological data available in this zone at synoptic stations like airports as shown on this map (white spots):

[Figure]

- Also, instead of showing a difference between two lead times in appendix, I would discuss the variability of forecast lead in terms of standard deviation. A standard deviation is appropriate to describe a variability.

**Minor comments**

The paper deserves to be carefully checked, there are many typo errors. Among them:

I would suggest to reformulate the title like "Variability of mineral dust forecast during the 2021 CADDIWA experiment"

I have reformulated the abstract like this:

"As an operational support to the CADDIWA field campaign, the coupled regional model WRF-CHIMERE is deployed in forecast mode during the summer 2021. The simulation domain covers the West Africa and the East Atlantic and allows the modeling of dust emissions and their transport to the Atlantic. On this route, we find Cape Verde which was used as a base for measurements during the CADDIWA campaign. Meteorological variables and mineral dust concentrations are forecasted on a horizontal grid with a 30 km resolution and from the surface to 200 hPa. Each day, the simulation starts the day before (D-1) and up to five days ahead (D+4). For each day, we thus have six different calculations, with expectantly a better precision the closer we get to the analysis (lead D-1). In this study, a quantification of the forecast variability of wind, temperature, precipitations and mineral dust concentrations according to the modelled lead is presented. It has been shown that the forecast quality

does not decrease with time and the high variability observed on some days for some variables (wind, temperature) does not explain the behavior of other dependent and downwind variables (mineral dust concentrations). A new method is also tested to create an ensemble without perturbing input data, but considering six forecast leads available for each date as members of an ensemble forecast. It has been shown that this new forecast based on this ensemble, is able to give better results for two AERONET stations on the four available for Aerosol Optical Depth. This could open the door to further testing with more complex operational systems."

Line 16: this sentence is strange "Leading the African continent to arrive above the Atlantic sea, they can generate tropical storms", please reformulate

Line 34: analyzed

Line 43: Change "And for those institutes that don't have the computer resources to do ensemble simulations" to "And for institutes which do not have sufficient computing resources to perform classical ensemble simulations"

Line 58: again five days?

Lin 76: I would say "The goal is not to perform a comparison of model with observations"

Line 135: diagnoses

Line 157: I am a bit surprised by the definition this normalized RMSE, can the author justify this formula?

Line 174: I would write : "desert- and a maritime-infuenced…"

L191-193: the last sentence needs to be reformulated

L201: what "model realizations" means?

L209: I would say "Having only si**x** members, once ranked from the lowest to the highest, this value is in fact the mean average of the 3th and 4th members"

L211: Through the paper harmonize the dates, 1 September and 31 September here, but check everywhere. In any case 31th is not correct.

L213 : realization?

L224: laos?

L254: Capo verde or Cape verde, check in the whole paper and harmonize

I suggest this conclusion:

"In this study, the first goal was to examine the variability of the forecast. This forecast was daily performed for six days, during the period August to October 2021 and as support for the CADDIWA field campaign. For meteorological variables (2m temperature, 10m wind speed, total precipitation rate) and surface concentrations of mineral dust, the day to day variability was quantified. Comparisons at two sites were performed, Bodélé (desert area and important source of dust) and Capo Verde (where the measurements of DACCIWA were coordinated). It has been shown that the wind speed is highly variable for day to day forecast while the temperature is stable over land but more variable over sea and shores (Capo Verde being a group of little Islands). The less stable parameter is

the precipitation at one location when the model may forecast an event one day and not at all the day after (and vice-versa).

Fits, one goal of the study was to examine if a large forecast variability at one site (such as Bodélé) may have a visible impact at a downwind remote site (such as Capo Verde). No evidence of a transport of variability (or a transport of stability) was found during the forecast. The large variability of wind speed, precipitation and temperature induce a large variability of the surface concentration of mineral dust. Between forecast leads, large differences were found both for the correlation and the bias. Considering the model configuration used for this study where no direct and indirect effects of aerosols on meteorology and only mineral dust as natural emissions was taken into account, this variability could be underestimated.

A next study could be to replay this forecast with a model version including all anthropogenic and natural emissions in the CHIMERE model with an exhaustive evaluation with the measurements of the experiment to come.

Second, a new way combining forecast leads was tested to improve the predictions. Considering that several forecast leads may be considered as the members of an ensemble, they are combined from (D-1) to (D+4) for all coinciding dates computing the mean and median values. These new "forecast leads" are compared, with all others members, to the Aerosol Optical Depth measurements of AERONET using correlation, nRMSE and bias statistics. It is noticeable that the forecast is not impaired when increasing the lead time. But it is also noticeable that out of four sites, the best scores for two sites are with the ensemble for the period of the CADDIWA campaign. It is not the case for an extended analyzed period, highlighting that the scores are close from one lead to another. The ensemble methodology provides the best scores when the AOD values are the most important and the most variables in time.

This result opens perspectives for forecasting in general. It would be interesting to test this hypothesis on operational systems: if the combination of the previous forecasts allows to improve the initial conditions of a new forecast, it would allow to perform less ensemble simulations for the same day and thus to reduce considerably the computing cost."

---

## Referee Comment (RC2)

[referee-annotated manuscript omitted]

---

## Author Comment (AC1)

Article gmd-2022-306:
**Variability and combination as ensemble of mineral dust forecast during the 2021 CAD-DIWA experiment**
L.Menut

**1    Answer to the Editor**

**Comments to the author:**

*Answer:*

Dear Editor and Reviewers,

Thanks for your interesting comments. The manuscript has evolved a lot since the first submission in January 2023. Following the editor's suggestions, simulations and analyses have been added to extend the period discussed. Following the reviewers' remarks, several paragraphs have been rewritten, statistical calculations have been revised and clarifications have been added.

Best regards,
Laurent Menut
May 22, 2023

**2    Answer to the Executive Editor**

Dear authors,
in my role as Executive editor of GMD, I would like to bring to your attention our Editorial version 1.2: https://www.geosci-model-dev.net/12/2215/2019/
This highlights some requirements of papers published in GMD, which is also available on the GMD website in the 'Manuscript Types' section: http://www.geoscientific-model-development.net/submission/manuscript_types.html
In particular, please note that for your paper, the following requirements have not been met in the Discussions paper:

- The main paper must give the model name and version number (or other unique identifier) in the title.
- If the model development relates to a single model then the model name and the version number must be included in the title of the paper. If the main intention of an article is to make a general (i.e. model independent) statement about the usefulness of a new development, but the usefulness is shown with the help of one specific model, the model name and version number must be stated in the title. The title could have a form such as, "Title outlining amazing generic advance: a case study with Model XXX (version Y)".

As you are using (WRF-)CHIMERE only, the model name including the version number you used has to appear in the title.
Yours, Astrid Kerkweg

*Answer:*
Dear Astrid Kerkweg, the article title was changed accordingly and is now: Variability and combination as ensemble of mineral dust forecast during the 2021 CADDIWA experiment, using the WRF 3.7.1 and CHIMERE v2020r3 models.

**3   Answers to the Reviewer 1**

This paper presents an evaluation of a forecast of mineral dust in support to the CADDIWA experiment over the western part of Africa. Also, an innovative way to produce an ensemble forecast is presented by using lead time of a single model instead of using several models. The coupled regional model WRF-CHIMERE is deployed in forecast mode during the summer 2021. I think this publication is suitable for publication after the following corrections.

**3.1   Major comments**

- My feeling is that the new methodology (using an ensemble of forecast lead) proposed in this paper could be applied to the general purpose of weather forecast. Since the WRF-CHIMERE also deliver data, I strongly suggest to apply the method to the main meteorological data available in this zone at synoptic stations like airports as shown on this map (white spots):

[Figure]

  *Answer:*
  Statistical scores calculated with meteorological data was added in the manuscript, following this suggestion. Data were extracted from the "University of Wyoming" meteorological dataset and scores were calculated in the same way that for the Aerosol Optical Depth. Two new tables of results and two time-series were added in the manuscript, section 4.

- Also, instead of showing a difference between two lead times in appendix, I would discuss the variability of forecast lead in terms of standard deviation. A standard deviation is appropriate to describe a variability.

  *Answer:*
  We agree that standard deviation is also appropriate to describe a variability. We used RMSE a lot in the study for the statistical scores of 2m temperature, 10m wind speed and AOD. For this specific case, in Appendix, we would like really to present differences maps for one date in order to see the instantaneous impact of the forecast on one specific date.

**3.2   Minor comments**

The paper deserves to be carefully checked, there are many typo errors. Among them:

- I would suggest to reformulate the title like "Variability of mineral dust forecast during the 2021 CADDIWA experiment"

  *Answer:*
  We think it is important to keep "combination as an ensemble" in the title since it is the added value as model development in this study.

[Figure]

Figure 1: *Time-series of 2m temperature (°C) and 10m wind speed (m.s⁻¹) measured and modelled during the forecast and for several leads. The last time-series, called ENSmean and ENSmedian, are the mean averaged and the median values of the previous leads, from (D-1) to (D+4).*

- I have reformulated the abstract like this:

  "As an operational support to the CADDIWA field campaign, the coupled regional model WRF-CHIMERE is deployed in forecast mode during the summer 2021. The simulation domain covers the West Africa and the East Atlantic and allows the modeling of dust emissions and their transport to the Atlantic. On this route, we find Cape Verde which was used as a base for measurements during the CADDIWA campaign. Meteorological variables and mineral dust concentrations are forecasted on a horizontal grid with a 30 km resolution and from the surface to 200 hPa. Each day, the simulation starts the day before (D-1) and up to five days ahead (D+4). For each day, we thus have six different calculations, with expectantly a better precision the closer we get to the analysis (lead D-1). In this study, a quantification of the forecast variability of wind, temperature, precipitations and mineral dust concentrations according to the modelled lead is presented. It has been shown that the forecast qualitydoes not decrease with time and the high variability observed on some days for some variables (wind, temperature) does not explain the behavior of other dependent and downwind variables (mineral dust concentrations). A new method is also tested to create an ensemble without perturbing input data, but considering six forecast leads available for each date as members of an ensemble forecast. It has been shown that this new forecast based on this ensemble, is able to give better results for two AERONET stations on the four available for Aerosol Optical Depth. This could open the door to further testing with more complex operational systems."

  *Answer:*
  OK, thank you, it is corrected.

- Line 16: this sentence is strange "Leading the African continent to arrive above the Atlantic sea, they can generate tropical storms", please reformulate

  *Answer:*
  Yes, sorry, it is a typo: "Leaving..."

- Line 34: analyzed

  *Answer:*
  corrected.

- Line 43: Change "And for those institutes that don't have the computer resources to do ensemble simulations" to "And for institutes which do not have sufficient computing resources to perform classical ensemble simulations"

  *Answer:*
  OK, corrected.

- Line 58: again five days?

  *Answer:*
  D-1, D+0, D+1, D+2, D+3 and D+4: six days.

- Lin 76: I would say "The goal is not to perform a comparison of model with observations"

  *Answer:*
  OK corrected.

- Line 135: diagnoses

  *Answer:*
  OK.

- Line 157: I am a bit surprised by the definition this normalized RMSE, can the author justify this formula?

  *Answer:*
  OK. To clarify the discussion, the definition was changed to use the "classical" Root Mean Square Error:

$$\text{RMSE} = \sqrt{\frac{1}{T}\sum_{t=1}^{T}\left(O_{t,i} - M_{t,i}\right)^2} \tag{1}$$

  All statistical scores were recalculated with this new formula.

- Line 174: I would write : "desert- and a maritime-infuenced..."

  *Answer:*
  OK corrected.

- L191-193: the last sentence needs to be reformulated

  *Answer:*
  OK, the new sentence is: "A common result to all parameters is that the best scores (for bias and correlation) are often obtained for lead time close to the analysis (D-1)."

- L201: what "model realizations" means?

  *Answer:*
  A model realization is one simulation with one specific set-up. It is currently used in articles dealing with ensemble modeling. Exemple: https://arxiv.org/pdf/1709.02798.pdf

- L209: I would say "Having only six members, once ranked from the lowest to the highest, this value is in fact the mean average of the 3th and 4th members"

  *Answer:*

- L211: Through the paper harmonize the dates, 1 September and 31 September here, but check everywhere. In any case 31 th is not correct.

  *Answer:*
  OK the manuscript was complete;ly checked for the date. I adopt the format: September 1, 2021.

- L213 : realization?

  *Answer:*
  See below

- L224: laos?

  *Answer:*
  Sorry, "also"

- L254: Capo verde or Cape verde, check in the whole paper and harmonize

  *Answer:*
  OK, the name is Cape Verde (the English version of the name).

- I suggest this conclusion:

  "In this study, the first goal was to examine the variability of the forecast. This forecast was daily performed for six days, during the period August to October 2021 and as support for the CADDIWA field campaign. For meteorological variables (2m temperature, 10m wind speed, total precipitation rate) and surface concentrations of mineral dust, the day to day variability was quantified. Comparisons at two sites were performed, Bodélé (desert area and important source of dust) and Capo Verde (where the measurements of DACCIWA were coordinated). It has been shown that the wind speed is highly variable for day to day forecast while the temperature is stable over land but more variable over sea and shores (Capo Verde being a group of little Islands). The less stable parameter is the precipitation at one location when the model may forecast an event one day and not at all the day after (and vice-versa).

  First, one goal of the study was to examine if a large forecast variability at one site (such as Bodélé) may have a visible impact at a downwind remote site (such as Capo Verde). No evidence of a transport of variability (or a transport of stability) was found during the forecast. The large variability of wind speed, precipitation and temperature induce a large variability of the surface concentration of mineral dust. Between forecast leads, large differences were found both for the correlation and the bias. Considering the model configuration used for this study where no direct

and indirect effects of aerosols on meteorology and only mineral dust as natural emissions was taken into account, this variability could be underestimated. A next study could be to replay this forecast with a model version including all anthropogenic and natural emissions in the CHIMERE model with an exhaustive evaluation with the measurements of the experiment to come.

Second, a new way combining forecast leads was tested to improve the predictions. Considering that several forecast leads may be considered as the members of an ensemble, they are combined from (D-1) to (D+4) for all coinciding dates computing the mean and median values. These new "forecast leads" are compared, with all others members, to the Aerosol Optical Depth measurements of AERONET using correlation, nRMSE and bias statistics. It is noticeable that the forecast is not impaired when increasing the lead time. But it is also noticeable that out of four sites, the best scores for two sites are with the ensemble for the period of the CADDIWA campaign. It is not the case for an extended analyzed period, highlighting that the scores are close from one lead to another. The ensemble methodology provides the best scores when the AOD values are the most important and the most variables in time. This result opens perspectives for forecasting in general. It would be interesting to test this hypothesis on operational systems: if the combination of the previous forecasts allows to improve the initial conditions of a new forecast, it would allow to perform less ensemble simulations for the same day and thus to reduce considerably the computing cost."

*Answer:*
The conclusion was updated with these suggestions.

**4    Answers to the Reviewer 2**

The paper presents the outcome of the forecasting tool implemented to accompagny the CADDIWA field campaign in September 2021. The forecasts were made using a simplified version of the WFR-CHIMERE model in order for the model products to be available each morning to the scientists in Cape Verde. The model outputs are compared to AERONET stations derived aerosol optical depths, when/where such measurements are available. Besides data/model comparisons, the paper develops interesting ideas on how to use the 6-days forecasts made daily in an "ensemble" mode. Metrics are provided that illustrate the insights that can be gained with the new approach. Even though, the approach is only tested on a limited number of variables and locations, the paper deserves to be published... Additional assessments on other variables than AOD would be a plus.

*Answer:*
I sinceraly acknowledge the Reviewer for the time spent on the manuscript and all proposed corrections. For the last remark, we agree and this will be the focus on the next study including all possible emissions (fires, dust, sea salt etc.) and the online coupling between WRF and CHIMERE to add the direct and indirect of aerosols on radiation and cloud formation. But, for this specific study, focusing on the variability of dust emissions and transport only, the AOD remains the only variable that I can really compare to the simulations.

Nevertheless, the english is quite rough and needs to be worked on. I have attached a copy of the typescript with suggestions on how to improve the quality of the english.

*Answer:*
The attached PDF file contains both English corrections and questions. The questions, inserted as little yellow boxes, are listed below. The English corrections are done directly into the new manuscript version and are readable with the differences PDF file.

- p.1 l.20: Tons of US references that need to be cited, the most recent ones at least.

*Answer:*
The two following references were added, representative of the sentence: (Martinez and Chaboureau, 2018) and (Price et al., 2018).

- p.1 l.25: this is for Aerolus... How about the other missions?

  *Answer:*
  Some references were also added here for EarthCare and IASI: (Clerbaux et al., 2009) and (Illingworth et al., 2015)

- p.2 l.30: Why different meteorology? different initial conditions ..? imposed by fresh analysis/re-analysis?

  *Answer:*
  Each day, the meteorological forecast is different. Then initial conditions are different for each simulation, as well as the complete meteorology.

- p.2 l.47: correlation with what?

  *Answer:*
  The sentence was changed as: "Second to try to establish some correlations between possible differences in forecast results."

- p.2 l.51: No other comparisons to AERONET data such as water vapor?

  *Answer:*
  It is a really good suggestion. I d'on't have these data and never tried to analyze them, but for sure in a next study.

- p.3 l.65: Add more reference of operational centers

  *Answer:*
  The new sentence is: "Note that the CHIMERE model is also used daily in forecast mode for air quality with all available chemical processes, being operated by operational centers such as Prevair or Copernicus, (Rouïl et al., 2009; Marécal et al., 2015)." There is indeed other forecast performed with this model, but only the two cited ones are really 'operational'.

- Table 1: specify countries associated with the AERONET stations

  *Answer:*
  It was added in the table which is now:

| Station Name | $\lambda$ (°E) | $\phi$ (°N) | AERONET | UWYO |
|---|---|---|---|---|
| Bodélé (Tchad) | 15.5 | 16.5 | x | |
| Zinder (Niger) | 8.98 | 13.75 | x | x |
| Banizoumbou (Niger) | 2.66 | 13.54 | x | |
| Niamey (Niger) | 2.2 | 16.43 | x | x |
| Cinzana (Mali) | -5.93 | 13.28 | x | |
| Dakar (Senegal) | -17.36 | 14.75 | x | x |
| CapeVerde (Cabo Verde) | -22.95 | 16.75 | x | x |

Table 1: *List of the AERONET and meteorological UWYO sites used for the comparisons between measured and modelled AOD, 2m temperature and 10m wind speed. Informations are the longitude $\lambda$, latitude $\phi$ for each site.*

- p.6 l.130: botton line? so what happens on the 11 September? Why is this a remarkable day across West Africa?

  *Answer:*
  Because, except, the meteorological situation, and as described a few lines before (l.124), it is a day when the forecast particularly underestimate the mineral dust concentrations when the forecast was several days in advance, but not just before the event, with (D+0) and (D+1).

- p.7 l.137: ? unclear

  *Answer:*
  It was rewritten as: "diagnose a larger wind speed then a faster transport"

- p.8 l.155 as in Aeronet stations?

  *Answer:*
  The formulas are general, and, yes, this could be Aeronet stations.

- Table 2: are you considering an area? rather than a grid point? if yes how big is the area around the selected points?

  *Answer:*
  The model information is extracted using the four grid points around the station by using a bilinear interpolation.

- p.11 l.200: What about Niamey and Dakar that are introduced in one of your previous table?

  *Answer:*

- p.13 l.246: be more precise

  *Answer:*
  The new sentence is "In this study, the first goal was to examine the variability of the forecast as a function of the lead time and for each forecasted day."

**References**

Clerbaux, C., Boynard, A., Clarisse, L., George, M., Hadji-Lazaro, J., Herbin, H., Hurtmans, D., Pommier, M., Razavi, A., Turquety, S., Wespes, C., and Coheur, P.-F.: Monitoring of atmospheric composition using the thermal infrared IASI/MetOp sounder, Atmospheric Chemistry and Physics, 9, 6041–6054, doi: 10.5194/acp-9-6041-2009, 2009.

Illingworth, A. J., Barker, H. W., Beljaars, A., Ceccaldi, M., Chepfer, H., Clerbaux, N., Cole, J., Delanoé, J., Domenech, C., Donovan, D. P., Fukuda, S., Hirakata, M., Hogan, R. J., Huenerbein, A., Kollias, P., Kubota, T., Nakajima, T., Nakajima, T. Y., Nishizawa, T., Ohno, Y., Okamoto, H., Oki, R., Sato, K., Satoh, M., Shephard, M. W., Velazquez-Blazquez, A., Wandinger, U., Wehr, T., and van Zadelhoff, G.-J.: The EarthCARE Satellite: The Next Step Forward in Global Measurements of Clouds, Aerosols, Precipitation, and Radiation, Bulletin of the American Meteorological Society, 96, 1311 – 1332, doi: https://doi.org/10.1175/BAMS-D-12-00227.1, 2015.

Marécal, V., Peuch, V.-H., Andersson, C., Andersson, S., Arteta, J., Beekmann, M., Benedictow, A., Bergström, R., Bessagnet, B., Cansado, A., Chéroux, F., Colette, A., Coman, A., Curier, R. L., Denier van der Gon, H. A. C., Drouin, A., Elbern, H., Emili, E., Engelen, R. J., Eskes, H. J., Foret, G., Friese, E., Gauss, M., Giannaros, C., Guth, J., Joly, M., Jaumouillé, E., Josse, B., Kadygrov, N., Kaiser, J. W., Krajsek, K., Kuenen, J., Kumar, U., Liora, N., Lopez, E., Malherbe, L., Martinez, I., Melas, D., Meleux, F., Menut, L., Moinat, P., Morales, T., Parmentier, J., Piacentini, A., Plu, M.,

Poupkou, A., Queguiner, S., Robertson, L., Rouïl, L., Schaap, M., Segers, A., Sofiev, M., Tarasson, L., Thomas, M., Timmermans, R., Valdebenito, A., van Velthoven, P., van Versendaal, R., Vira, J., and Ung, A.: A regional air quality forecasting system over Europe: the MACC-II daily ensemble production, Geoscientific Model Development, 8, 2777–2813, doi: 10.5194/gmd-8-2777-2015, 2015.

Martinez, I. R. and Chaboureau, J.-P.: Precipitation and Mesoscale Convective Systems: Radiative Impact of Dust over Northern Africa, Monthly Weather Review, 146, 3011 – 3029, doi: https://doi.org/10.1175/MWR-D-18-0103.1, 2018.

Price, H. C., Baustian, K. J., McQuaid, J. B., Blyth, A., Bower, K. N., Choularton, T., Cotton, R. J., Cui, Z., Field, P. R., Gallagher, M., Hawker, R., Merrington, A., Miltenberger, A., Neely III, R. R., Parker, S. T., Rosenberg, P. D., Taylor, J. W., Trembath, J., Vergara-Temprado, J., Whale, T. F., Wilson, T. W., Young, G., and Murray, B. J.: Atmospheric Ice-Nucleating Particles in the Dusty Tropical Atlantic, Journal of Geophysical Research: Atmospheres, 123, 2175–2193, doi: https://doi.org/10.1002/2017JD027560, 2018.

Rouïl, L., Honoré, C., Vautard, R., Beekmann, M., Bessagnet, B., Malherbe, L., Meleux, F., Dufour, A., Elichegaray, C., Flaud, J., Menut, L., Martin, D., Peuch, A., Peuch, V., and Poisson, N.: PREV'AIR : an operational forecasting and mapping system for air quality in Europe, Bull Am Meteorol Soc, 90, 73–83, doi: 10.1175/2008BAMS2390.1, 2009.